Provenance-and machine learning-based recommendation of parameter values in scientific workflows

Silva Junior Daniel 1 danieljunior@id.uff.br
Pacitti Esther 2
http://orcid.org/0000-0002-9089-7303 Paes Aline 1
http://orcid.org/0000-0001-9346-7651 de Oliveira Daniel 1
1 Institute of Computing, Universidade Federal Fluminense , Niteroi, RJ , Brazil
2 Inria, CNRS, LIRMM, University of Montpellier , Montpellier , France
Ardagna Claudio
Electronic publication date: 2021 Jul 5
Publication date: 2021
Volume: 7
Electronic Location ID: e606
Received 2020 Nov 11; Accepted 2021 May 31
Copyright: © 2021 Silva Junior et al.
Copyright year: 2021
Copyright holder: Silva Junior et al.
License: This is an open access article distributed under the terms of the Creative Commons Attribution License, which permits unrestricted use, distribution, reproduction and adaptation in any medium and for any purpose provided that it is properly attributed. For attribution, the original author(s), title, publication source (PeerJ Computer Science) and either DOI or URL of the article must be cited.
License URL: https://creativecommons.org/licenses/by/4.0/

Keywords: Scientific workflows, Recommender systems, Machine Learning, Preference Learning

Funding: CNPq, FAPERJ and CAPES Finance Code 001 This work was supported by the Brazilian research agencies CNPq, FAPERJ, and CAPES (Finance Code 001). The funders had no role in study design, data collection and analysis, decision to publish, or preparation of the manuscript.

==============================
Scientific Workflows (SWfs) have revolutionized how scientists in various domains of science conduct their experiments. The management of SWfs is performed by complex tools that provide support for workflow composition, monitoring, execution, capturing, and storage of the data generated during execution. In some cases, they also provide components to ease the visualization and analysis of the generated data. During the workflow’s composition phase, programs must be selected to perform the activities defined in the workflow specification. These programs often require additional parameters that serve to adjust the program’s behavior according to the experiment’s goals. Consequently, workflows commonly have many parameters to be manually configured, encompassing even more than one hundred in many cases. Wrongly parameters’ values choosing can lead to crash workflows executions or provide undesired results. As the execution of data- and compute-intensive workflows is commonly performed in a high-performance computing environment e.g., (a cluster, a supercomputer, or a public cloud), an unsuccessful execution configures a waste of time and resources. In this article, we present FReeP—Feature Recommender from Preferences, a parameter value recommendation method that is designed to suggest values for workflow parameters, taking into account past user preferences. FReeP is based on Machine Learning techniques, particularly in Preference Learning. FReeP is composed of three algorithms, where two of them aim at recommending the value for one parameter at a time, and the third makes recommendations for n parameters at once. The experimental results obtained with provenance data from two broadly used workflows showed FReeP usefulness in the recommendation of values for one parameter. Furthermore, the results indicate the potential of FReeP to recommend values for n parameters in scientific workflows.

Introduction

Scientific experiments are the basis for evolution in several areas of human knowledge (De Oliveira, Liu & Pacitti, 2019; Mattoso et al., 2010; Hey & Trefethen, 2020; Hey, Gannon & Pinkelman, 2012). Based on observations of open problems in their research areas, scientists formulate hypotheses to explain and solve those problems (Gonçalves & Porto, 2015). Such hypothesis may be confirmed or refuted, and also can lead to new hypotheses. For a long time, scientific experiments were manually conducted by scientists, including instrumentation, configuration and management of the environment, annotation and analysis of results. Despite the advances obtained with this approach, time and resources were wasted since a small misconfiguration of the parameters of the experiment could compromise the whole experiment. The analysis of errors in the results was also far from trivial (De Oliveira, Liu & Pacitti, 2019).

The evolution in computer science field allowed for the development of technologies that provided useful support for scientists in their experiments. One of these technologies is Scientific Workflows (De Oliveira, Liu & Pacitti, 2019; Deelman et al., 2005). A scientific workflow (named workflow henceforth) is an abstraction that represents each step of the experiment expressed as an activity, which has input data and relationships (i.e., data dependencies) with other activities, according to the stages of the experiment (Zhao & Ioan Raicu, 2008).

Several workflows commonly require the execution of multiple data-intensive operations as loading, transformation, and aggregation (Mattoso et al., 2010). Multiple computational paradigms can be used for the design and execution of workflows, e.g., shell and Python scripts (Marozzo, Talia & Trunfio, 2013), Big Data frameworks (e.g., Hadoop and Spark) (Guedes et al., 2020b), but they are usually managed by complex engines named Workflow Management Systems (WfMS). A key feature that a WfMS must address is the efficient and automatic management of parallel processing activities in High Performance Computing (HPC) environments (Ogasawara et al., 2011). Besides managing the execution of the workflow in HPC environments, WfMSs are also responsible for capturing, structuring and recording metadata associated to all the data generated during the execution: input data, intermediate data, and the final results. These metadata is well-known as provenance (Freire et al., 2008). Based on provenance data, it is possible to analyze the results obtained and to foster the reproducibility of the experiment, which is essential to prove the veracity of a produced result.

In this article, the concept of an experiment is seen as encompassing the concept of a workflow, and not as a synonym. A workflow may be seen as a controlled action of the experiment. Hence, the workflow is defined as one of the trials conducted in the context of an experiment. In each trial, the scientist needs to define the parameter values for each activity of the workflow. It is not unusual that a simple workflow has more than 100 parameters to set. Setting up these parameters may be simple for an expert, but not so simple for non-expert users. Although WfMSs represent a step forward by providing the necessary infrastructure to manage workflow executions, they provide a little help (or even no help at all) on defining parameter values for a specific workflow execution. A good parameters values tune in a workflow execution is crucial not only for the quality of the results but also influences if a workflow will execute or not (avoiding unnecessary execution crashes). A poor choice of parameters values can cause failures, which leads to a waste of execution time. Failures caused by poor choices of parameter values are even more severe when workflows are executing in HPC environments that follow a pay-as-you-go model, e.g., clouds, since they can increase the overall financial cost.

This way, if the WfMS could “learn” from previous successfully executions of the workflow and recommend parameter values for scientists, some failures could be avoided. This recommendation is especially useful for non-expert users. Let us take as an example a scenario where an expert user has modeled a workflow and executed several trials of the same workflow varying the parameter values. If a non-expert scientist wants to execute the same workflow with a new set of parameter values and input data, but does not know how to set the values of some of the parameters, one can benefit from parameter values used on previous executions of the same (or similar) workflow. The advantage of the WfMS is provenance data already contains the parameter values used on previous (successful) executions and can be a rich resource to be used for recommendation. Thus, this article hypothesis is that by adopting an approach to recommend the parameters values of workflows in a WfMS, we can increase the probability that the execution of workflow will be completed. As a consequence, the financial cost associated with execution failures is reduced.

In this article, we propose a method named FReeP—Feature Recommender From Preferences, which aims at recommending values for parameters of workflow activities. The proposed approach is able to recommend parameter values in two ways: (i) a single parameter value at a time, and (ii) multiple parameter values at once. The proposed approach relies on user preferences, defined for a subset of workflow parameters, together with the provenance of the workflow. It is essential to highlight that user preferences are fundamental to explore experiment variations in a scientific scenario. Furthermore, for our approach, user preferences help prune search space and consider user restrictions, making personalized recommendations. The idea of combining user preferences and provenance is novel and allows for producing a personalized recommendation for scientists. FReeP is based on Machine Learning algorithms (Mitchell, 2015), particularly, Preference Learning (Fürnkranz & Hüllermeier, 2011), and Recommender Systems (Ricci, Rokach & Shapira, 2011). We evaluated FReeP using real workflow traces (considered as benchmarks): Montage (Hoffa et al., 2008) from astronomy domain and SciPhy (Ocaña et al., 2011) from bioinformatics domain. Results indicate the potential of the proposed approach. This article is an extension of the conference paper “FReeP: towards parameter recommendation in scientific workflows using preference learning” (Silva Junior et al., 2018) published in the Proceedings of the 2018 Brazilian Symposium on Databases (SBBD). This extended version provides new empirical shreds of evidence regarding several workflow case studies as well as a broader discussion on related work and experiments.

This article is organized in five sections besides this introduction. “Background” section details the theoretical concepts used in the proposal development. “FReeP—Feature Recommender from Preferences” section presents the algorithm developed for the problem of parameters value recommendation using user preferences. “Experimental Evaluation” section shows the results of the experimental evaluation of the approach in three different scenarios. Then, “Related Work” section presents a literature review with papers that have addressed solutions to problems related to the recommendation applied to workflows and the Machine Learning model hyperparameter recommendation. Lastly, “Conclusion” section brings conclusions about this article and points out future work.

Background

This section presents key concepts for understanding the approach presented in this article to recommend values for parameters in workflows based on users’ preferences and previous executions. Initially, it is explained about scientific experiments. Following, the concepts related to Recommender Systems are presented. Next, the concept of Preference Learning is presented. This section also brings a Borda Count overview, a non-common voting schema that is used to decide which values to suggest to the user.

Scientific experiment

A scientific experiment arises from the observation of some phenomena and questions raised from the observation. The next step is the hypotheses formulation aiming at developing possible answers to those questions. Then, it is necessary to test the hypothesis to verify if an output produced is a possible solution. The whole process includes many iterations of refinement, consisting, for example, of testing the hypothesis under distinct conditions, until it is possible to have enough elements to support it.

The scientific experiment life-cycle proposed by Mattoso et al. (2010) is divided into three major phases: composition, execution and analysis. The composition phase is where the experiment is designed and structured. Execution is the phase where all the necessary instrumentation for the accomplishment of the experiment must be finished. Instrumentation means the definition of input data, parameters to be used at each stage of the experiment, and monitoring mechanisms. Finally, the analysis phase is when the data generated by the composition and execution phases are studied to understand the obtained results. The approach presented in this article focus on the Execution phase.

Scientific workflows

Scientific workflows have become a de facto standard for modeling in silico experiments (Zhou et al., 2018). A Workflow is an abstraction that represents the steps of an experiment and the dataflow through each of these steps. A workflow is formally defined as a directed acyclic graph W(A,Dep). The nodes A={a1,a2,…,an} are the activities and the edges Dep represent the data dependencies among activities in A. Thus, given ai :amp:mid; (1 ≤ i ≤ n), the set P = {p1, p2, …, pm} represents the possible input parameters for activity ai that defines the behavior of ai. Therefore, a workflow can be represented as a graph where the vertices act as experiment steps and the edges are the relations, or the dataflow between the steps.

A workflow can also be categorized according to the level of abstraction into conceptual or concrete. A conceptual workflow represents the highest level of abstraction, where the experiment is defined in terms of steps and dataflow between them. This definition does not explain how each step of the experiment will execute. The concrete workflow is an abstraction where the activities are represented by the computer programs that will execute them. The execution of an activity of the workflow is called an activation (De Oliveira et al., 2010b), and each activation invokes a program that has its parameters defined. However, managing this execution, which involves setting the correct parameter values for each program, capturing the intermediate data and execution results, becomes a challenge. It was with this in mind, and with the help of the composition of the experiment in the workflow format, that Workflow Management Systems (WfMS), such as Kepler (Altintas et al., 2006), Pegasus (Deelman et al., 2005) and SciCumulus (De Oliveira et al., 2010b) emerged.

In special, SciCumulus is a key component of the proposed approach since it provides a framework for parallel workflows to benefit from FReeP. Also, data used in the experiments presented in this article are retrieved from previous executions of several workflows in SciCumulus. It is worth noticing that other WfMSs such as Pegasus and Kepler could also benefit from FReeP as long as they provide necessary provenance data for recommendation. SciCumulus architecture is modularized to foster maintainability and ease the development of new features. SciCumulus is open-source and can be obtained at https://github.com/UFFeScience/SciCumulus/. The system is developed using MPI library (a de facto standard library specification for message-passing), so SciCumulus is a distributed application, i.e., each SciCumulus module has multiple instances created on the machines of the distributed environment (which are different processes and each process has multiple threads) that communicate, triggering functions for sending and receiving messages between these processes. According to Guerine et al. (2019), SciCumulus has four main modules: (i) SCSetup, (ii) SCStarter, (iii) SCCore, and (iv) SCQP (SciCumulus Query Processor). The first step towards executing a workflow in SciCumulus is to define the workflow specification and the parameters values to be consumed. This is performed using the SCSetup module. The user has to inform the structure of the workflow, which programs are associated to which activities, etc. When the metadata related to the experiment is loaded into the SciCumulus database, the user can start executing the workflow. Since SciCumulus was developed focusing on supporting the execution of workflows in clouds, instantiating the environment was a top priority. The SCSetup module queries the provenance database to retrieve prospective provenance and creates the virtual machines (in the cloud) or reserve machines (in a cluster). The SCStarter copies and invokes an instance of SCCore in each machine of the environment, and since SCCore is a MPI-based application it runs in all machines simultaneously and follows a Master/Worker architecture (similar to Hadoop and Spark). The SCCore-Master (SCCore0) schedules the activations for several workers and each worker has a specific ID (SCCore1, SCCore2, etc.). When a worker is idle, it sends a message for the SCCore0 (Master) and request more activations to execute. The SCCore0 defines at runtime the best activation to send following a specific cost model. The SCQP component allows for users to submit queries to the provenance database for runtime or post-mortem analysis. For more information about SciCumulus please refer to (De Oliveira et al., 2012; De Oliveira et al., 2010a; Guerine et al., 2019; Silva et al., 2020; Guedes et al., 2020a; De Oliveira et al., 2013).

Provenance

An workflow activation has input data, and generates intermediate and output data. WfMS has to collect all metadata associated to the execution in order to foster reproducibility. This metadata is called provenance (Freire et al., 2008). According to Goble (2002), the provenance must verify data quality, path audit, assignment verification, and information querying. Data quality check is also related to verifying the reliability of workflow generated data. Path audit is the ability to follow the steps taken at each stage of the experiment that generated a given result. The assignment verification is linked to the ability to know who is responsible for the data generated. Lastly, an information query is essential to analyze the data generated by the experiment’s execution. Especially for workflows, provenance can be classified as prospective (p-prov) and retrospective (r-prov) (Freire et al., 2008). p-prov represents the specification of the workflow that will be executed. It corresponds to the steps to be followed to achieve a result. r-prov is given by executed activities and information about the environment used to produce a data product, consisting of a structured and detailed history of the execution of the workflow.

Provenance is fundamental for the scientific experiment analysis phase. It allows for verifying what caused an activation to fail or generated an unexpected result, or in the case of success, what were the steps and parameters used until the result. Another advantage of provenance is the reproducibility of an experiment, which is essential for the validation of the results obtained by third parties. Considering the provenance benefits in scientific experiments, it was necessary to define a model of representation of provenance (Bose, Foster & Moreau, 2006). The standard W3C model is PROV (Gil et al., 2013). PROV is a generic data model and is based on three basic components and their links, being the components: Entity, Agent and Activity. The provenance and provenance data model are essential concepts because FReeP operation relies on provenance to recommend parameter values. Also, to extract provenance data to use in FReeP it is necessary to understand the provenance data model used.

Recommender systems

FReeP is a personalized Recommender system (RS) (Resnick & Varian, 1997) aiming at suggesting the most relevant parameters to the user to perform a task, based on their preferences.There are three essential elements for the development of a recommender system: Users, Items, and Transactions. The Users are the target audience of the recommender system with their characteristics and goals. Items are the recommendation objects and Transactions are records that hold a tuple (user, interaction), where the interaction encompasses the actions that the user performed when using the recommender system. These interactions are generally user feedbacks, which may be interpreted as their preferences.

A recommender task can be defined as: given the elements Items, Users and Transactions, find the most useful items for each user. According to Adomavicius & Tuzhilin (2005), a recommender system must satisfy the equation ∀u∈U,iu′=argmaxi∈IF(u,i), where U represents the users, I represents the items and F is a utility function that calculates the utility of an item i in I for a u in U user. In case the tuple (u, i) is not defined in the entire search space, the recommender system can extrapolate the F function.

The utility function varies according to the approach followed by the recommender system. Thus, recommender systems are categorized according to the different strategies used to define the utility function. The most common approaches to recommender systems are: Content Based, Collaborative Filtering and Hybrids. Figure 1 provides a taxonomy related types of recommender systems for this work.

Figure 1 Related types of recommender systems taxonomy.

In Collaborative Filtering Recommender Systems, a recommendation is based on other users’ experience with items in the system domain. The idea is related to the human behavior of, at times, giving credit to another person’s opinion about what should be done in a given situation. The Neighborhood Based subtype strictly follows the principle that users with similar profiles have similar preferences. The Model-Based subtype generates a hypothesis from the data and use it to make recommendations instantly. Although widely adopted, Collaborative Filtering only uses collective information, limiting novel discoveries in scientific experiment procedures.

Content-based Recommender Systems make recommendations similar to items that the user has already expressed a positive rating in the past. To determine the similarity degree between items, this approach is highly dependent on extracting their characteristics. However, each scenario needs the right item representation to give satisfactory results. In scientific experiments, it can be challenging to find an optimal item representation.

Finally, Hybrid Recommender Systems arise out of an attempt to minimize the weaknesses that traditional recommendation techniques have when used individually. Also, it is expected that a hybrid strategy can aggregate the strengths of the techniques used together. There are several methods of combining recommendation techniques in creating a hybrid recommender system, including: Weighting approaches that provides a score for each recommendation item, Switching, which allows for selecting different types of recommending strategies, Mixing, to make more than one recommentation at a time, Feature Combination, to put together both Content-Based and Collaborative Filtering strategies, Cascade, that first filters the candidate items for the recommendation, followed by refining these candidates, looking for the best alternatives, Feature Augmentationand Meta-Level, which chain a series of recommendations one after another (Burke, 2002).

FReeP is as a Cascade Hybrid Recommender System because the content of user preferences is used to prune the search space followed by a collaborative strategy to give the final recommendations.

Preference learning

User preferences play a crucial role in recommender systems (Viappiani & Boutilier, 2009). From an Artificial Intelligence perspective, a preference is a problem restriction that allows for some degree of relaxation. Fürnkranz & Hüllermeier (2011) refers to Learning Preferences as “inducing preference models from empirical data”. In several scenarios, the empirical data is implicitly defined, for example, when the user’s preference is expressed by clicking on the most interesting products, instead of effectively buying one of them or stating that one is preferable over another.

A Preference Learning task consists of learning a predictive function that, given a set of items where preferences are already known, predicts preferences for a new set of items. The most common way of representing preferences is through binary relationships. For example, a tuple (xi > xj) > would mean a preference for the value i over j for the attribute x.

The main task within Preference Learning area is Learning to Rank as commonly it is necessary to have an ordering of the preferences. The task is divided into three categories: Label Ranking (Vembu & Gärtner, 2011), Instance Ranking (Bergeron et al., 2008) and Object Ranking (Nie et al., 2005). In Label ranking a ranker makes an ordering of the set of classes of a problem for each instance of the problem. In cases where the classes of a problem are naturally ordered, the instance ranking task is more suitable, as it orders the instances of a problem according to their classes. The instances belonging to the “highest” classes precede the instances that belong to the “lower” classes. In object ranking an instance is not related to a class. This task’s objective is, given a subset of items referring to the total set of items, to produce a ranking of the objects in that subset—for example, the ranking of web pages by a search engine.

Pairwise Label Ranking (Fürnkranz & Hüllermeier, 2003; Hüllermeier et al., 2008) (PLR) relates each instance with a preference type a > b, representing that a is preferable to b. Then, a binary classification task is assembled where each example a, b is annotated with a is a is preferable over b and 0, otherwise. Then, a classifier M a, b is trained over such dataset to learn how to make the preference predictions which returns 1 as a prediction that a is preferable to b and 0 otherwise. Instead of using a single classifier that makes predictions between m classes, given a set L of m classes, there will be m(m − 1)/2 binary classifiers, where a classifier M i, j only predicts between classes i, j in L. Then, the strategy defined by PLR uses the prediction of each classifier as a vote and uses a voting system that defines an ordered list of preferences. Next, we give more details about how FReeP tackles the voting problem.

Borda count

Voting Theory (Taylor & Pacelli, 2008) is an area of Mathematics aimed at the study of voting systems. In an election between two elements, it is fair to follow the majority criterion, that is, the winning candidate is the one that has obtained more than half of the votes. However, elections involving more than two candidates require a more robust system. Preferential Voting (Karvonen, 2004) and Borda Count (Emerson, 2013) are two voting schemas concerning the scenarios where there are more than two candidates. In Preferential Voting, voters elicit a list of the most preferred to the least preferred candidate. The elected candidate is the one most often chosen as the most preferred by voters.

Borda Count is a voting system in which voters draw up a list of candidates arranged according to their preference. Then, each position in the user’s preference list gets a score. In a list of n candidates, the candidate in the i-th position on the list receives the score n − i. To determine the winner, the final score is the sum of each candidate’s scores for each voter, and the candidate with the highest score is the elected one.

Figure 2 depicts an example of Borda Count. There are four candidates: A, B, C and D, and five vote ballots. The lines in each ballot represent the preference positions occupied by each candidate. As there are four candidates, the candidate preferred by a voter receives three points. The score for the candidate D is computed as follows: 1 voter elected the candidate D as the preferred candidate, then 1 * 3 = 3 points; 2 voters elected the candidate D as the second most preferred candidate, then 2 * 2 = 4 points; 2 voters elected the candidate D as the third most preferred candidate, then 2 * 1 = 2 points; 0 voters elected the candidate D as the least preferred, then 0 * 0 = 0 points. Finally, candidate D total score = 3 + 4 + 2 + 0 = 9.

Figure 2 Votes example that each candidate received in voters preference order.

Voting algorithms are used together with recommender systems to choose which items the users have liked best to make a good recommendation. Rani, Shokeen & Mullick (2017) proposed a recommendation algorithm based on clustering and a voting schema that after clustering and selecting the target user’s cluster, uses the Borda Count to select the most popular items in the cluster to be recommended. Similarly, Lestari, Adji & Permanasari (2018) compares Borda Count and the Copeland Score Al-Sharrah (2010) in a recommendation system based on Collaborative Filtering. Still using the Borda Count, Tang & Tong (2016) proposes the BordaRank. The method consists of using the Borda Count method directly in the sparse matrix of evaluations, without predictions, to make a recommendation.

FReeP—Feature Recommender from Preferences

Figure 3 depicts a synthetic workflow, where one can see four activities represented by colored circles where activities 1, 2, and 3 have one parameter each. To execute the workflow, it is required to define values for parameters 1, 2, and 3. Given a scenario where a user has not defined values for all parameters, FReeP targets at helping the user to define values for the missing parameters. For this, FReeP divides the problem into two sub-tasks: (1) recommendation for only one parameter at a time; (2) recommendation for n parameters at once. The second task is more challenging than the first as parameters of different activities may present some data dependencies.

Figure 3 A synthetic workflow: circles represent activities, arrows between the circles represent the link between activities (data dependencies), and the labels for each circle represent the configuration parameters for each activity.

Taking into account user preferences, FReeP suggests parameter values that maximize a probability to make the workflow execute flawlessly until its end. FReeP receives a user preferences set to yield the personalized recommendations. The recommendations are the output of a model induced by a Machine Learning technique. FReeP is a hybrid recommendation technique as it incorporates aspects of both Collaborative Filtering and Content-Based concepts.

The way FReeP tackles the recommendation task is presented in three versions. In the first two versions, the algorithm aims at recommending a value for only one parameter at a time. While the naive version assumes that all parameters have a discrete domain, the enhanced second version is an extension of the first one that is able to deal with cases where a parameter has a continuous domain. The third version targets at recommending values for n > 1 parameters at a time.

Next, we start by presenting the naive version of the method that makes the recommendation for a single parameter at a time. Then, we follow to the improved version with enhancements that improve the performance and allows for working with parameters in the continuous domain. Finally, a generic version of the algorithm is presented, aiming at making the recommendation of values for multiple parameters at a time.

Discrete domain parameter value recommendation

Given a provenance database D, a parameter y ∈ Y, where Y is the workflow parameters set, and a preferences or restrictions set P defined by the user, where pi ∈ P (yi, valk), FReeP one parameter approach aims at solving the problem of recommending a r value for y, so that the P preferences together with the r recommendation to y maximize the chances of the workflow activation to run to the end.

Figure 4 presents an architecture overview of FReeP’s naive version. The algorithm receives as input the provenance database, a target workflow and user preferences. User preferences are also input as this article assumes that the user already has a subset of parameters for which has already defined values to use. In this naive version, the user preferences are only allowed in the form a = b, where a is a parameter, and b is a desired value to a.

Figure 4 FReeP architecture overview.

Based on the user’s preferences, it would be possible to query the provenance database from which the experiment came from to retrieve records that could assist in the search for other parameters values that had no preferences defined. However, FReeP is based on a model generation that generalizes the provenance database, removing the user’s need to perform this query yet providing results that the query would not be able to return.

To obtain a recommendation from FReeP’s naive version, seven steps are required: partitions generation, horizontal filter, vertical filter, hypothesis generation, predictions, aggregation, and, finally election-based recommendation. Algorithm 1 shows the proposed algorithm to perform the parameter recommendation, considering the preferences for a subset or all other workflow execution parameters.

The algorithm input data are: target parameter for which the algorithm should make the recommendation, y; user preferences set, such as a list of key-values, where the key is a workflow parameter and value is the user’s preference for that parameter, P; provenance database, D.

The storage of provenance data for an experiment may vary from one WfMS to another. For example, SciCumulus, which uses a provenance representation derived from PROV, stores provenance in a relational database. Using SciCumulus example, it is trivial for the user responsible for the experiment to elaborate a SQL query that returns the provenance data related to the parameters used in each activity in a key-value representation. The key-value representation can be easily stored in a csv format file, which is the required format expected as provenance dataset in FReeP implementation. Thus, converting provenance data to the csv format is up to the user. Still, regarding the provenance data, the records present in the algorithm input data containing information about the parameters must be related only to executions that were successfully concluded, that is, there was no failure that resulted in the execution abortion. The inclusion of components to query and transform provenance data and force successful executions parameters selection would require implementations for each type of WfMS, which is out of the scope of this article.

The initial step, partitions_generation, builds partitioning rules set based on the user’s preferences. Initially, the preference set parameters P are used to generate a powerset. This first step returns all generated powerset as a partitions ruleset. Figure 5 shows an example of how this first step works, with some parameters from SciPhy workflow.

Figure 5 Example of FReeP’s partitioning rules generation for Sciphy provenance dataset using user’s preferences.

Then, FReeP initializes an iteration over the partitioning rules generated by the previous step. Iteration begins selecting only the records that follow the user’s preferences contained in the current ruleset, named in the algorithm as horizontal_filter. Figure 6 uses the partitions presented in Fig. 5 to show how the horizontal_filter step works.

Figure 6 Example of FReeP’s horizontal filter using one partitioning rule for the Sciphy provenance dataset.

Subsequently, in the vertical_filter step, there is a parameter removal that aims at keeping only the recommendation target parameter, the parameters present in the current set of partitioning rules, and those that are neither the recommendation target parameter nor are present in any of the original user preferences. The last parameters mentioned remain because, in a next step, they can help to build a more consistent model. Thus, let PW be all workflow parameters set; PP the workflow parameters for which preference values have been defined; PA the parameters present in the partitioning rules of an iteration over the partitioning rules and PV = (PW − PP) ∪(PP ∩ PA) ∪ {y} ; the output from vertical_filter is the data from horizontal_filter for parameters in PV. Figure 7 uses data from the examples in Figs. 5 and 6 to show how the vertical_filter step works.

Figure 7 FReeP’s vertical filter step.

The chain comprising the partitions generation and the horizontal and vertical filters is crucial to minimize the Cold Start problem (Lika, Kolomvatsos & Hadjiefthymiades, 2014). Cold Start is caused by the lack of ideal operating conditions for an algorithm, specifically in the recommender systems. This problem occurs, for example, when there are few users for the neighborhood definition with a similar user profile or lack of ratings for enough items. FReeP can also be affected by Cold Start problem. If only all preferences were used at one time for partitioning the provenance data, in some cases, it could be observed that the resulting partition would be empty. This is because there could be an absence of any of the user’s preferences in the provenance data. Therefore, generating multiple partitions with subsets of preferences decreases the chance of obtaining only empty partitions. However, in the worst case where none of the user’s preferences are present in the workflow provenance, FReeP will not perform properly, thus failing to make any recommendations.

After the partitions generation and horizontal and vertical filters are discovered, there is a filtered data set that follows part of the user’s preferences. These provenance data that will generate the Machine Learning model have numerical and categorical domain parameters. However, traditional Machine Learning models generally work with numerical data because the generation of these models, in most cases, involves many numerical calculations. Therefore, it is necessary to codify these categorical parameters to a numerical representation. The technique used here to encode categorical domain parameters to numerical representation is One-Hot encoding (Coates & Ng, 2011). This technique consists of creating a new binary attribute, that is, the domain of this new attribute is 0 or 1, for each different attribute value present in dataset.

The encoded provenance data allows building Machine Learning models to make predictions for the target parameter under the step hypothesis_generation. The model generated has the parameter y as class variable, and the other parameters present in vertical_filter step output data are the attributes used to generalize the hypothesis. The model can be a classifier, where the model’s prediction is a single recommendation value, or a ranker, where its prediction is an ordered list of values, of the value most suitable for the recommendation to the least suitable.

With a model created, we can use it to recommend the value for the target parameter. This step is represented in FReeP as recommended, and the recommendation of parameter y is made from the user’s preferences. It is important to emphasize that the model’s training data may contain parameters that the user did not specify any preference. In this case, an attribute of the instance submitted for the hypothesis does not have a defined value. To clarify the problem, let PW be all workflow parameters set, PP the parameters of workflow for which preference values have been defined; PA the parameters present in the partition rules of an iteration over the partitioning rules; and PV = (PW − PP) ∪ (PP ∩ PA) ∪ {y}, there may be parameters p ∈ PV | p ∉ PP, and for those parameters p there are no values defined a priori. To handle this problem, the average values present in the provenance data are used to fill in the numerical attributes’ values and the most frequent values in the provenance date for the categorical attributes.

All predictions generated by recommend step, which is within the iteration over the partitioning rules, are stored. The last algorithm step, elect_recommendation, uses all of these predictions as votes to define which value should be recommended for the target parameter. When an algorithm instance is setup to return a classifier type model in hypothesis_generation step, the most voted value is elected as the recommendation. On the other hand, when an algorithm instance is setup to return a ranker type model in hypothesis_generation step, the strategy is Borda Count. The use of the Borda Count strategy seeks to take advantage of the list of lists form that the saved votes acquire when using the ranker model. This list of lists format occurs because the ranker prediction is a list, and since there are as many predictions as partitioning rules, the storage of these predictions takes the list of lists format.

Discrete and continuous domain parameter value recommendation

The naive version of FReeP allowed evaluating the algorithm’s proposal. The proposal showed relevant results after initial tests (presented in next section), so efforts were focused on improving its performance and utility. In particular, the following problems have been identified: (1) User has some restriction to set his/her parameters preferences; (2) The categorical domain parameters when used as a class variable (parameters for recommendation) are treated as well as they are present in the input data; (3) Machine Learning models used can only learn when the class (parameter) variable has a discrete domain; (4) All partitions generated by workflow parameters powerset present in user preferences are used as partitioning rules for the algorithm.

Regarding problem 1, in Algorithm 1, the user was limited to define his preferences with the equality operator. Depending on the user’s preferences, the equality operator is not enough. With this in mind, the Enhanced FReeP allows for the user to have access to the relational operators: ==,>,>=,<,<= and != to define his/her preferences. In addition, two logical operators are also supported in setting preferences: | and &. Preferences with combination of supported operators is also allowed, for example: (a > 10) | (at < 5).

Algorithm 1 Naive FReeP-Discrete.

Require:	
y:recommendationtargetparameter	
P:{(param,val)|paramisaworkflowparameter,valisthepreferencevalueforparam}	
D:{{(param11,val11),...,(param1l,val1l),...(paramlm,vallm)}|listheworkflowparametersnumber,mistheprovenancedatasetlength}	
1: procedure FReeP(y, P, D)	
2: partitions ← partitions_generation(P, D)	
3: votes ← ø	
4: for each partition ∈ partitions do	
5: data ← horizontal_filter(D, partition)	
6: data ← vertical_filter(data, partition)	
7: data ← ′ preprocessing(data)	
8: model ← hypothesis_generation(data′, y)	
9: vote ← recommend(model, y)	
10: votes ← votes ∪ {vote}	
11: recommendation ← elect_recommendation(votes)	
12: return recommendation	

However, by allowing users to define their preferences in this way we create a problem when setting up the instances for recommendation step. As seen, PW represents all workflow parameters set, PP are workflow parameters that preference values have been set; PA the parameters present in the partitioning rules of an iteration over the partition rules; and PV = (PW − PP) ∪(PP ∩ PA) ∪ {y}. Thus, there may be parameters p ∈ PV | p ∉ PP, and for those parameters p, there are no values defined a priori. This enhanced version of the proposal allows the user’s preferences to be expressed in a more relaxed way, demanding to create the instances used in the step recommendation that include a range (or set of values). To handle this isse, all possible instances from preference values combinations were generated. In case the preference is related to a numerical domain parameter and is defined in terms of values range, like a ≤ 10.5, FReeP uses all values present in the source provenance database that follows the preference restriction. It is important to note that for both numerical and categorical parameters, the combination of possible values are those present in the provenance database and that respect the user’s preferences. Then, predictions are made for a set of instances using the model learned during the training phase.

Regarding problem 2, the provenance database, in general, present attributes with numerical and categorical domains. It is FReeP responsibility to convert categorical values into numerical representation due to restrictions related to the nature of the training algorithms of the Machine Learning models, e.g., Support Vector Machines (SVM) (Wang, 2005).

This pre-processing step was included in Algorithm 2 as classes_preprocessing step. The preprocessing consists in exchanging each distinct categorical value for a distinct integer. Note that the encoding of the parameter used as a class variable in the model generation is different from the encoding applied to the parameters used as attributes represented by the step preprocessing.

Algorithm 2 Enhanced FReeP.

Require:	
y:recommendationtargetparameter	
P:{(param,val)|paramisaworkflowparameter,valisthepreferencevalueforparam}	
D:{{(param11,val11),...,(param1l,val1l),...(paramlm,vallm)}|listheworkflowparametersnumber,mistheprovenancedatasetlength}	
1: procedure FReeP(y, P, D)	
2: D′ ← classes_preprocessing(D)	
3: partitions ← optimized_partitions_generation(P, D′)	
4: votes ← ø	
5: for each partitionpartition ∈ partitions do	
6: data ← horizontal_filter(D′, partition)	
7: data ← vertical_filter(data, partition)	
8: model_type ←model_select(data, y)	
9: data′ ← preprocessing(data)	
10: model ← hypothesis_generation(data′, y, model_type)	
11: vote ← recommend(model, y)	
12: votes ← votes ∪ {vote }	
13: recommendation ← elect_recommendation(votes)	
14: return recommendation	

Concerning problem 3, by using classifiers to handle a continuous domain class variable degrades the performance results. Performance degradation happens because the numerical class variables are considered as categorical. For continuous numerical domain class variables, the Machine Learning models suggested are Regressors (Myers & Myers, 1990). In this way, the Enhanced FReeP checks the parameter y domain, which is the recommendation target parameter, represented as model_select step in Algorithm 2.

To analyze problem 4, it is important to note that after converting categorical attributes One-Hot encoding in preprocessing step, the provenance database will have a considerable increase in the number of attributes. Also, after categorical attributes encoding in preprocessing step, the parameters extracted from the user’s preferences, are also encoded for partitions_generation step. In Algorithm 1, the partitioning rules powerset is calculated on all attributes derived from the original parameters after One-Hot encoding. If FReeP uses the powerset generated from the parameters present in the user’s preferences set as partitioning rules (in the partitions_generation step), it can be very costly. Thus, using the powerset makes the complexity of the algorithm becomes exponential according to the parameters present in the user’s preferences set. Alternatives to select the best partitioning rules and handle the exponential cost are represented in Algorithm 2 as optimized_partitions_generation step. The two strategies proposed here are based on Principal Components Analysis (PCA) (Garthwaite et al., 2002) and the Analysis of variance (ANOVA) (Girden, 1992) statistical metric.

The strategy based on PCA consists of extracting x principal components from all provenance database, pcaD, and for each pt ∈ partitions, pcapti, which are pt partition principal components. Then, the norms are calculated ||pcaD − pcapti||, and from that n partitioning rules are selected that generated pcapti such that ||pcaD − pcapti|| resulted in the lowest calculated values. Note that both x and n are defined parameters when executing the algorithm. In summary, the PCA strategy will select the partitions where the main components extracted are the closest to the principal components of the original provenance dataset.

ANOVA strategy seeks the n partitioning rules that best represent D, selecting those that generate partitions where the data variance is closest to D data variance. In short, original data variance and data variance for each partition are calculated using the ANOVA metric, then partitions with most similar variance to the original provenance data are selected. Here, the n rules are defined in terms of the data percentage required to represent the entire data set, and that parameter must also be defined in algorithm execution. Using PCA or ANOVA partitioning strategies means that the partitioning rules used by FReeP can be reduced, depending on the associated parameters that need to be defined.

Recommendation for n Parameters at a time

Algorithms 1 and 2 aim at producing single parameter recommendation at a time. However, in a real usage scenario of scientific workflows, the WfMS will probably need to recommend more than one parameter at a time. A naive alternative to handle this problem is to execute Algorithm 2 for each of the target parameters, always adding the last recommendation to the user’s preference set. This alternative assumes that the parameters to be recommended are independent random variables. One way to implement this strategy is by using a classifiers chain (Read et al., 2011).

Nevertheless, this naive approach neglects that the order in which the target parameters are used during algorithm interactions can influence the produced recommendations. The influence is due to parameter dependencies that can be found between two (or more) workflow activities (e.g., two activities consume a parameter produced by a third activity of the workflow). In Fig. 3, the circles represent the activities of workflow, so activities 2 and 3 are preceded by activity 1 (e.g., they consume the output of activity 1). Using this example, we can see that it is possible that there is a dependency relationship between the parameters param2 and param3 with the parameter param1. In this case, the values of param2 and param3 parameters can be influenced by parameter param1 value.

In order to deal with this problem, FReeP leverages the Classifiers Chains Set (Read et al., 2011) concept. This technique allows for estimating the joint probability distribution of random variables based on a Classifiers Chains Set. In this case, the random variables are the parameters for which values are to be recommended, and the joint probability distribution concerns the possible dependencies between these parameters. The Classifiers Chains and Classifiers Chains Set are techniques from Multi-label Classification (Tsoumakas & Katakis, 2007) Machine Learning task.

Figure 8 depicts an architecture overview for the proposed algorithm named as Generic FReeP that recommends n parameters simultaneously. The architecture presented in Fig. 8 shows that the solution developed to make n parameter recommendations at a time is a packaging of FReeP algorithm to one parameter. This final approach is divided into five steps: identification of parameters for the recommendation, generation of ordered sequences of these parameters, iteration over each of the sequences generated with the addition of each recommendation from FReeP to the user preferences set, separation of recommendations by parameter and finally the choice of value recommendation for each target parameters. The formalization can be seen in Algorithm 3.

Figure 8 Generic FReeP architecture overview.

Algorithm 3 Generic FReeP.

Require:	
P:recommendationtargetparameter	
D:{(param,val)|paramisaworkflowparameter,valisthepreferencevalueforparam}	
D:{{(param11,val11),...,(param1l,val1l),...(paramlm,vallm)}|listheworkflowparametersnumber,mistheprovenancedatasetlength}	
N :number of random sequences orders to be generated	
1: procedure Generic FReeP(P, D, N)	
2: target_parameters ← parameters_extractor(P, D)	
3: votes ← ø	
4: for each param ∈ target_parameters do	
5: votes ← votes ∪ {(param, []) }	
6: ordered_sequences ← sequence_generator(target_parameters, N)	
7: for each sequence ∈ ordered_sequences do	
8: preferences_tmp ← P	
9: for each param ∈ sequence do	
10: recommendation ← FReeP(param, preferences_tmp, D)	
11: votes[param] ← votes[param] ∪ recommendation	
12: new_preference ← generate_preference(param, recommendation)	
13: preferences_tmp ← preferences_tmp ∪ new_preference	
14: response ← ø	
15: for each (param, values) ∈ votes do	
16: response[param] most_voted(values)	
17: return response	

The first step parameters_extractor extracts the workflow parameters that are not present in the users’ preferences and will be the targets of the recommendations. Thus, all other parameters that are not in the user’s preferences will have recommendation values.

Lines 4 and 5 of the algorithm comprise the initialization of the variable responsible for storing the different recommendations for each parameter during the algorithm execution. Then, the list of all parameters that will be recommended is used for generating different ordering of these parameters, indicated by sequence_generators step. For example, let w be a workflow with 4 p parameters and let u be an user with pr1 and pr3 preferences for the p1 and p3 parameters respectively. The parameters to be recommended are p2 and p4, in this case two possible orderings are: {p2, p4} and {p4, p2}. Note that the number of sorts used in the algorithm are not all possible sorts, in fact N of the possible sorts are selected at random.

Then, the algorithm initializes an iteration over each of the sorts generated by the step sequence_generators. Another nested iteration over each parameter present in the current order also begins. An intuitive explanation of the algorithm between lines 9 and 13 is that each current sequence parameter is used together with the user’s preferences for its recommendation. At the end of the recommendation of one of the ordering parameters, the recommendation is incorporated into the preferences set used in the recommendation of the next ordering parameter. In this iteration, the recommendations are grouped by parameter to facilitate the election of the recommended value for each target parameter.

The step of iterating over the generated sequences, always adding the last recommendation to the set of preferences, is the Classifiers Chains concept. To deal with the dependency between the workflow parameters that can influence a parameter value recommendation, the step that generates multiple sequences of parameters, combined with the Classifiers Chains, is the Classifiers Chains Set concept.

Finally, to choose the recommendation for each target parameter, a vote is taken on lines 15 and 16. The most_voted procedure makes the majority election that defines the target parameter recommendation value. This section presented three algorithms that are part of the FReep approach developed for the parameter recommendation problem in workflows. The proposals covered two main scenarios for parameters value recommendation (single and multiple parameter at a time).

Experimental Evaluation

This section presents the experimental evaluation of all versions of FReeP. First, we present the workflows used as case studies namely SciPhy (Ocaña et al., 2011) and Montage (Jacob et al., 2009). Following we present the experimental and environment setups. Finally, we discuss the results.

Case studies

In this article, we consider two workflows from bioinformatics and astronomy domains, namely SciPhy (Ocaña et al., 2011) and Montage (Jacob et al., 2009), respectively. SciPhy is a phylogenetic analysis workflow that generates phylogenetic trees (a tree-based representation of the evolutionary relationships among organisms) from input DNA, RNA and aminoacid sequences. SciPhy has four major activities as presented in Fig. 9A: (i) sequence alignment, (ii) alignment conversion, (iii) evolutionary model election and (iv) tree generation. SciPhy has been used in scientific gateways such as BioInfoPortal (Ocaña et al., 2020). SciPhy is a CPU-intensive workflow, bacause many of its activities (especially the evolutionary model election) commonly execute for several hours depending on the input data and the chosen execution environment.

Figure 9 The abstract specification of (a) SciPhy and (b) Montage.

Montage (Jacob et al., 2009) is a well-known astronomy workflow that assembles astronomical images into mosaics by using FITS (Flexible Image Transport System) files. Those files include a coordinate system and the image size, rotation, and WCS (World Coordinate System) map projection. Figure 9B shows the montage activities: (i) ListFITS, which extracts compressed FITS files, (ii) Projection, which maps the astronomical positions into a Euclidean plane, (iii) SelectProjections, which joins the planes into a single mosaic file, and (iv) CreateIncorrectedMosaic, which creates an overlapping mosaic as an image. Programs (v) CalculateOverlap, (vi) ExtractDifferences, (vii) CalculateDifferences, (viii) FitPlane, and (ix) CreateMosaic refine the image into the final mosaic. Montage is a data-intensive workflow, since one single execution of Montage can produce several GBs of data.

Experimental and environment setup

All FReeP algorithms presented in this article were implemented using the Python programming language. FReeP implementation also benefits from Scikit-Learn (Pedregosa et al., 2011) to learn and evaluate the Machine Learning models, numpy (Van der Walt, Colbert & Varoquaux, 2011), a numerical data manipulation library; and pandas (McKinney, 2011), which provides tabular data functionalities.

The machine specification where experiments were performed is a CPU Celeron (R) Dual-Core T3300 @ 2.00 GHz × 2 processor, 4GB DDR2 RAM and 132 GB HDD. To measure recommendations performance when the parameter is categorical, precision and recall are used as metrics. Precision and recall are metrics widely used for the quantitative assessment of recommender systems (Herlocker et al., 2004; Schein et al., 2002). Eq. (1) defines precision and Eq. (2) defines recall, following the recommender vocabulary, where TR is the correct recommendation set and R is all recommendations set. An intuitive explanation to precision is that it represents the most appropriate recommendations fraction. Still, recall represents the appropriate recommendation fraction that was made.

(1) precision=∥TR∩R∥∥R∥

(2) recall=∥TR∩R∥∥TR∥

(3) MSE=1n∑i=1n(RV−TV)2

When the parameter to be recommended is numerical, the performance of FReeP is evaluated with Mean Square Error (MSE). The MSE formula is given by Eq. (3) where n is the recommendations number, TV is the correct recommendation values set, and RV is the recommended values set.

Dataset

The datasets used are provenance data extracted from real executions of the workflows SciPhy (all executions) using SciCumulus Workflow Management System and Montage (part of the executions with SciCumulus) and part of the execution data gathered at the Workflow Generator site (https://confluence.pegasus.isi.edu/display/pegasus/WorkflowGenerator). This site provides instances of real workflow for evaluation of algorithms and systems on a range of workflow sizes. All data within these workflow traces is gathered from real executions of scientific workflows on the grid and in the cloud from the Pegasus’ team in ISI at the University of Southern California. The SciPhy executions consumed from 200 up to 500 fasta files downloaded from RefSeq database. The Montage executions consumed from 50 up to 100 FIT files obtained from the “Two Micron All-Sky Survey”1 . In the case of SciPhy, the executions were performed by 3 different users (one expert and 2 undergraduate students). In the case of Montage (the executions in SciCumulus were performed by an undergraduate student and the ones downloaded from the Workflow Generator site were performed by experts).

Table 1 summarizes the main characteristics of the datasets. The Total Records column shows the number of past executions of each workflow. Each dataset record can be used as an example for generating Machine Learning models during the algorithm’s execution. As seen, the SciPhy dataset is relatively small compared to Montage. The column Total Attributes shows how many activity parameters are considered in each workflow execution. Both workflows have the same number of categorical domain parameters, as presented in the column Categorical Attributes. Montage has more numeric domain parameters than SciPhy, as shown in the Numerical Attributes column.

Table 1 Dataset characteristics.

Dataset	Total records	Total attributes	Categorical attributes	Numerical attributes	
Sciphy	376	6	2	4	
Montage	1,565	8	2	6	

Statistics on the SciPhy numerical attributes are shown in Table 2. This table presents the minimum and maximum values of each attribute, in addition to the standard deviation. The attribute prob1 (probability of a given evolutive relationship is valid) has the highest standard deviation, and its range of values is the largest among all attributes. The prob2 attribute (probability of a given evolutive relationship is valid) has both a range of values and the standard deviation similar to prob1. The standard deviation of the values of num_aligns (total number of alignments in a given data file) is very small, while the attribute length (maximum sequence length in a specific data file) has a high standard deviation, considering its values range.

Table 2 SciPhy dataset statistics.

Parameter	Minimum value	Maximum value	Standard deviation	
num_aligns	9.00	11.00	0.21	
length	85.00	1,039.00	169.90	
prob1	634.67	5,753.52	1,103.43	
prob2	635.87	5,795.28	1,101.76	

The Montage numerical attributes, shown in Table 3, in most of the cases, have smaller standard deviation than the SciPhy. On average, Montage attributes also have a smaller values range than SciPhy dataset attributes. Also, in Montage dataset, the crota2 attribute (a float value that represents an image rotation on sky) has the largest values range and the largest standard deviation. The dec (an optional float value that represents Dec for region statistics) and crval2 (a float value that represents Axis 2 sky reference value in Montage workflow) attributes have close statistics and are the attributes with the smallest data range and the smallest Montage data standard deviation.

Table 3 Montage dataset statistics.

Parameter	Minimum value	Maximum value	Standard deviation	
cntr	0.00	134.00	35.34	
ra	83.12	323.90	91.13	
dec	−27.17	28.85	17.90	
crval1	83.12	323.90	91.13	
crval2	−27.17	28.85	17.90	
crota2	0.00	360.00	178.64	

In Fig. 10, it is possible to check the correlation between the different attributes in the datasets. It is notable in both Figs. 10A and 10B that the attributes (i.e., workflow parameters) present a weak correlation. All those statistics are relevant to understand the results obtained by the experiments performed from each version of FReeP algorithm.

Figure 10 Datasets attributes correlation matrices.

Discrete domain recommendation evaluation

This experiment was modeled to evaluate FReeP’s algorithm key concepts using the naive version presented in Algorithm 1, that was developed to recommend one discrete domain parameter at a time. This experiment aims at evaluating and comparing the performance of FReeP when its hypothesis_generation step instantiates either a single classifier or a ranker. The ranker tested as a model was implemented using the Pairwise Label Ranking technique. K Nearest Neighbors (Keller, Gray & Givens, 1985) classifier is used as the classifier of this ranker implementation. The k parameter of K Nearest Neighbors classifier was set as 3, 5, 7 for both the ranker and classifier. The choice of k ∈ {3,5,7} is because small datasets are used, and thus k values greater than 7 do not return any neighbors in the experiments.

Experiment 1. Algorithm 1 evaluation script

The algorithm is instantiated with the classifier or ranker and a recommendation target workflow parameter.

The provenance database is divided into k parts to follow a K-Fold Cross Validation procedure (Kohavi, 1995). At each step, the procedure takes k − 1 parts to train the model and the 1 remaining part to make the predictions. In this experiment, k = 5.

Each workflow parameter is used as recommendation target parameter.

Each provenance record in test data is used to retrieve target parameter real value.

Parameters that are not the recommendation target are used as preferences, with values from current test record.

Then, algorithm performs recommendation and both the result and the value present in the test record for the recommendation target parameter are stored.

Precision and recall values are calculated based on all K-Fold Cross Validation iterations.

Results

Experiment 1 results are presented and analyzed based on the values of precision and recall, in addition to the execution time. Figure 11 shows that Algorithm 1 execution with Sciphy provenance database, using both the classifier and the ranker. Only KNN classifier with k = 3 gives a precision greater than 50%. Also, a high standard deviation is noticed. Even with unsatisfactory performance, Fig. 12 shows that KNN classifier presented better recall results than those for precision, both in absolute values terms and standard deviation, which had a slight decrease. In contrast, the ranker recall was even worse with the precision results and still present a very high standard deviation.

Figure 11 Precision results with SciPhy data.

Figure 12 Recall results with SciPhy data.

Figure 13 shows the execution time, in seconds, to obtain the experiment’s recommendations for SciPhy. The execution time of ranker is much more significant when compared to the time spent by the classifier. This behavior can be explained by the fact that the technique used to generate the ranker creates multiple binary classifiers. Another point to note is that the execution time standard deviation from ranker is also very high. It is important to note that when FReeP uses KNN, it is memory-based, since each recommendation needs to be loaded into main memory.

Figure 13 Experiment recommendation execution time with SciPhy data.

Analyzing Fig. 14 (Montage) one can conclude that with the use of k = 3 for the classifier and for the ranker produces relevant results. The precision for this case reached 80%, and the standard deviation was considerably smaller compared to the precision results with Sciphy dataset in Fig. 11. For k ∈ {5,7}, the same results behavior was observed, considerably below those expected.

Figure 14 Precision results with Montage data.

Considering the precision, Fig. 15 shows that the results for k = 3 were the best for both the classifier and for ranker, although for this case they did not reach 80% (although it is close). It can be noted that the standard deviation was smaller when compared to the standard deviations found for precision. One interesting point about the execution time of the experiment with Montage presented in Fig. 16 is that for k ∈ {3,7} the ranker spent less time than the classifier. This behavior can be explained because the ranker, despite being generated by a process where several classifiers are built, relies on binary classifiers. When used alone, the classifier needs to handle all class variables values, in this case, parameter recommendation values, at once. However, it is also important to note that the standard deviation for ranker is much higher than for the classifier.

Figure 15 Recall results with Montage data.

Figure 16 Experiment recommendation execution time with Montage data.

In general, it was possible to notice that the use of ranker did not bring encouraging results. In all cases, ranker precision and recall were lower than those presented by the classifier. Besides, the standard deviation of ranker in the execution time spent results was also very high. Another point to be noted is that the best precision and recall results were obtained with the data from Montage workflow. These results may be linked to the fact that the Montage dataset has more records than the Sciphy dataset.

Discrete and continuous domain recommendation evaluation

Experiment 1 was modified to evaluate the Algorithm 2 performance, yielding Experiment 2. Algorithm 2 was executed with variations in the choice of classifiers and regressors, partitions strategies, and records percentage from provenance database. All values per algorithm parameter are presented in Table 4.

Table 4 Algorithm 2 values per parameter used in Experiment 2.

Classifiers	Regressors	Partition strategy	Percentage	
KNN	Linear Regression	PCA	30	
SVM	KNR	ANOVA	50	
Multi-Layer Perceptron	SVR		70	
	Multi-Layer Perceptron			

Experiment 2. Algorithm 2 evaluation script

Algorithm 2 is instantiated with a classifier or regressor, a partitioning strategy, percentage data to be returned by partitioning strategy, and a target workflow parameter.

Provenance database is divided using K-Fold Cross Validation, k = 5

Each provenance record on test data is used to retrieve the target parameter’s real value.

A random number x between 2 and parameters number present in provenance database is chosen to simulated preference number used in recommending target parameter.

x parameters are chosen from the remaining test record to be used as preferences.

Algorithm performs recommendation, and both result and test record value for the target parameter are stored.

Precision and recall, or MSE values are calculated based on all K-Fold Cross Validation iterations.

Results

Experiment 2 results are presented using precision, recall, and execution time for categorical domain parameters recommendations, while numerical domain parameters recommendations are evaluated using MSE and the execution time. Based on the results obtained in Experiment 1, only classifiers were used as Machine Learning models in Experiment 2, i.e., we do not consider rankers.

The first observation when analyzing the precision data in Fig. 17 is that ANOVA partitioning strategy obtained better results than PCA. ANOVA partitioning strategy precision in absolute values is generally more significant, and variation in precision for each attribute considered for recommendation is lower than PCA strategy. The classifiers have very similar performance for all percentages of partitions in the ANOVA strategy. On the other hand, the variation in the percentages of elements per partition also reflects a more significant variation in results between the different classifiers. The Multi Layer Perceptron (MLP) classifier, which was trained using the Stochastic Descending Gradient (Bottou, 2010) with a single hidden layer, presents the worst results except in the setup that it follows the PCA partitioning strategy with a percentage of 70% elements in the partitioning. The MLP model performance degradation may be related to the fact that the numerical attributes are not normalized before algorithm execution.

Figure 17 Precision results with Sciphy data.

Recall results, in Fig. 18 were very similar to precision results in absolute values. A difference is the smallest variation, in general, of recall results for each attribute used in the recommendation experiment. The Multi Layer Perceptron classifier presented a behavior similar to the precision results, with a degradation in the setup that includes ANOVA partitioning with 70% of the elements in the partitioning.

Figure 18 Recall results with Sciphy data.

Figure 19 shows the average execution time in seconds during the experiment with categorical domain parameters in each setup used. Execution time of ANOVA partitioning strategy was, on average, half the time used with the PCA partitioning strategy. The execution time using different classifiers for each attribute is also much smaller and stable for ANOVA strategy than for PCA, regardless of element partition percentage.

Figure 19 Experiment recommendation execution time with Sciphy data.

Analyzing precision, recall, and execution time spent data jointly, ANOVA partitioning strategy showed the best recommendation performance for the categorical domain parameters of the Sciphy provenance database. Going further, the element partition percentage generated by the strategy has no significant impact on the results. Another interesting point is that a simpler classifier like KNN presented results very similar to those obtained by a more complex classifier like SVM.

Figure 20A brings the data from results obtained for the numerical domain parameter Sciphy provenance database. The data shows zero MSE in all cases, except for the use of Multi Layer Perceptron in the regression. This result can be explained by the small database and the few different values for each numerical domain parameter. Small values difference per parameter suggests that the regressors have no work to generate a result equal to what is already present in the database.

Figure 20 MSE results and recommendation execution time with Sciphy data.

Looking at Fig. 20B, one can notice that, similar to the categorical domain parameters results, the execution time of ANOVA partitioning strategy is much less than the time used by the PCA strategy. Another similar point with categorical domain parameter results is the smaller and more stable ANOVA strategy results variation.

From all results obtained in the Experiment 2 using Sciphy provenance database, it can be noticed that the ANOVA partitioning strategy had the best performance. Further precision, recall, and MSE results, for the Algorithm 2 setup with ANOVA partitioning strategy also proved to be the one that performed the recommendations in the shortest time, generally in half the time that the PCA partitioning strategy. Note that the recommendation time can be treated as training time since the proposed algorithm has a memory-based approach. Finally, the choice of the generated partition size and the classifier or regressor used have no significant impact on the final result unless the classifier or regressor is based on Multi Layer Perceptron with the same parametrization used in this article.

Analyzing Fig. 21, precision results obtained with categorical domain parameters from Montage workflow provenance database is observed that in almost all the experiment setup variations evaluated, maximum performance is reached. As seen in Table 1, the Montage workflow provenance database used in the experiments has only two categorical domain parameters. The small variation in possible values in the database is an explanation for the precision results. The recall results in Fig. 22 are similar to the precision ones.

Figure 21 Precision results with Montage data.

Figure 22 Recall results with Montage data.

Concerning the results about the experiment time with categorical domain parameters from the Montage provenance database, presented in Fig. 23, one can see that the KNN classifier, k = 3, with PCA partitioning strategy was the most time-consuming. On the other hand, with the same PCA partitioning strategy, the Multi Layer Perceptron classifier used less time, but with a wide variation in recommendation times for different parameters. The ANOVA partitioning strategy continued to be a partitioning strategy that delivers the fattest recommendations. Still analyzing ANOVA partitioning strategy results, it is possible to see that the KNN classifier, with k ∈ {5,7}, was the fastest in recommending Montage workflow categorical domain parameters.

Figure 23 Experiment recommendation execution time with Montage data.

Making a general analysis of results in Figs. 21, 22 and 23, the setup that uses ANOVA partitioning strategy with the KNN classifier, k = 7 it’s the best. This setup was the one that obtained the best results for precision, recall, and execution time spent simultaneously. MSE results for Montage numerical domain parameters presented in Fig. 24A show that, in general, the MSE was very close to zero for all cases, except in algorithm setup using PCA partitioning strategy with 30% elements in the generated partition and the regressor implemented by Multi Layer Perceptron. The MSE and its variation were very close to zero.

Figure 24 MSE results and recommendation execution time with Montage data.

Regarding the execution time of Experiment 2 for numerical domain parameters recommendations for Montage data, Fig. 24B indicates the same behavior shown by results with SciPhy provenance database. Using ANOVA partitioning strategy and KNR regressors with k ∈ {5,7} as setup for Algorithm 2 produced the fastest recommendations.

The experiment execution time of Montage provenance database was much greater than the time used with the data from the workflow Sciphy. The explanation is the difference in the database size. Another observation is that the ANOVA partitioning strategy produces the fastest recommendations. Another point is that the percentage of the elements in partitioning generated by each partitioning strategy has no impact on the algorithm performance. Finally, it was possible to notice that the more robust classifiers and regressors had their performance exceeded by simpler models in some cases for the data used.

Generic FReeP recommendation evaluation

A third experiment was modeled to evaluate Algorithm 3 performance. As in Experiment 2, different variations, following Table 4 values, were used in algorithm execution. Precision, recall, and MSE are also the metrics used to evaluate the recommendations made by each algorithm instance.

Experiment 3. Algorithm 3 evaluation script

n Records from the provenance database were chosen as random examples.

m ≥ 2 random parameters were chosen for each example record as preferences, and their values are the same as those present in the example record.

Algorithm 3 was instantiated with a classifier or a regressor, a partitioning strategy, the partitions percentage to be returned by the partitioning strategy, and the selected m preferences.

Each returned recommendation is separated into numeric and categorical and is stored.

Precision and recall values were calculated for categorical recommendations and Mean Square Error (MSE) for numerical recommendations.

Results

Results showed here were obtained by fixing parameter n = 10 in Experiment 3, and using only SciPhy provenance database. Based on Experiment 2 results, it was decided to use the ANOVA partitioning strategy with 50% recovering elements from the provenance database. This choice is because the ANOVA partitioning strategy was the one that obtained the best results in previous experiment. As the percentage of data recovered by the strategy was not an impacting factor in the results, an intermediate percentage used in the previous experiment is selected. In addition, only KNN, with k ∈ {5,7}, and SVM were kept as classifiers, whereas only KNR, with k ∈ {5,7}, and SVR was chosen as regressors. These choices are supported by, in general, are the ones that present the best precision, recall, and MSE results in Experiment 2.

Table 5 presents the results obtained with the Algorithm 3 instance variations. Each row in the table represents an Algorithm 3 instance setup. The column that draws the most attention is the Failures. What happens is that, for some cases, the algorithm was not able to carry out the recommendation together and therefore did not return any recommendations. It is important to remember that each algorithm setup was tested on a set with 10 records extracted randomly from the database. The random record selection process can select records in which parameter values can be present only in the selected record. For this experiment, the selected examples are removed from the dataset, and therefore there is no other record that allows the correct execution of the algorithm.

Table 5 Experiment 3 results with Sciphy dataset.

Classifier	Regressor	Partitioning strategy	MSE	Precision	Recall	Failures	
KNN 5	KNR 5	ANOVA 50	0.0	1.0	1.0	6	
KNN 5	KNR 7	ANOVA 50	0.0	1.0	1.0	6	
KNN 5	SVR	ANOVA 50	1.1075	1.0	1.0	6	
KNN 7	KNR 5	ANOVA 50	4,279.2240	1.0	1.0	5	
KNN 7	KNR 7	ANOVA 50	0.0	1.0	1.0	5	
KNN 7	SVR	ANOVA 50	0.444	1.0	1.0	5	
SVM	KNR 5	ANOVA 50	1,148.1876	0.75	0.75	6	
SVM	KNR 7	ANOVA 50	0.0	1.0	1.0	7	
SVM	SVR	ANOVA 50	0.0	1.0	1.0	7	

Analyzing Table 5 results, focusing on the column Failures and taking into account that 10 records were chosen for each setup, it is possible to verify that in most cases, the algorithm was not able to make recommendations. However, considering only the recommendations made, it can be seen that the algorithm had satisfactory results for the precision and recall metrics. The values presented for the MSE metric were mostly satisfactory, differing only in the configurations of lines 4 and 7, both using the regressor KNR with k = 5. Another point to note is that the algorithm had more problems to make recommendations when the SVM classifier was used. Furthermore, it is possible to note that algorithm setups with more sophisticated Machine Learning models such as SVM and SVR do not add performance to the algorithm, specifically for Sciphy provenance dataset used.

Related Work

Previous literature works had already relied on recommender systems to support scientific workflows. Moreover, hyperparameter tuning methods also have similar goals as paramater recommendation. Hyperparameters are variables that cannot be estimated directly from data, and, as a result, it is the user’s task to explore and define those values. Hyperparameter Optimization (HPO) is a research area that emerged to assist users in adjusting the hyperparameters of Machine Learning models in a non-ad-hoc manner (Yang & Shami, 2020). The well-defined processes resulting from research in the area may speed up the experimentation process and allow for reproducibility and fair comparison between models. Among the different methods of HPO, we can mention Decision Theory, Bayesian Optimization, Multi-fidelity Optimization, and Metaheuristic Algorithms.

Among the Decision Theory methods, the most used are Grid Search (Bergstra et al., 2011) and Random Search (Bergstra & Bengio, 2012). For both strategies, the user defines a list of values to be experimented for each hyperparameter. In Grid Search, the search for optimum values is given by experimenting the predefined values for the entire cartesian product. Random Search selects a sample for the hyperparameters to improve the execution time of the whole process. While the exponential search space of Grid Search may be impossible to complain, in Random Search there is the possibility that an optimal combination will not be explored. Also, the common problem between both approaches is that the dependencies between the hyperparameters are not taken into account. FReeP considers the possible dependencies between parameters by following the concept of classifier chains.

The Bayesian Optimization (Eggensperger et al., 2013) method optimizes the search space exploration using information from the previously tested hyperparameters to prune the non-promising combinations test. Despite using a surrogate model, the Bayesian optimization method still requires that the target model evaluation direct the search for the optimal hyperparameters. In a scenario of scientific workflows, it is very costly from the economical and runtime perspective to run an experiment, even more so to only evaluate a combination of parameter values. FReeP does not require any new workflow execution to recommend which values to use as it uses only data from past executions.

Multi-fidelity Algorithms (Zhang et al., 2016) also have the premise of balancing the time spent to search for hyperparameters. This kind of algorithm is based on successively evaluating hyperparameters in a subset search space. Those strategies follow similar motivations as the partitions generation of FReeP. However, in a scenario of scientific workflows, the Multi-fidelity algorithms still require workflow execution to evaluate combination quality.

The Metaheuristics Algorithms (Gogna & Tayal, 2013), based on the evolution of populations, use different forms of combinations of pre-existing populations in the hope of generating better populations at each generation. For hyperparameters tuning, hyperparameters with missing values are the population. Still, FReeP does not require any new execution of the workflow a priori to evaluate a recommendation given by the algorithm.

In general, the works that seek to assist scientists with some type of recommendation involving scientific workflow are focused on the composition phase. Zhou et al. (2018) uses a graph-based clustering technique to recommend workflows that can be reused in the composition of a developing workflow. De Oliveira et al. (2008) uses workflow provenance to extract connection patterns between components in order to make recommendations of new components for a workflow in composition. For each new component used in the composition of workflow, new components are recommended. Halioui, Valtchev & Diallo (2016), uses Natural Language Processing combined with specific ontologies in the field of Bioinformatics to extract concrete workflows from works in the literature. After the reconstruction of concrete workflows, tool combinations patterns, its parameters, and input data used in these workflows are extracted. All this data extracted can be used as assistance for composing new ones workflows that solve problems related to the mined workflows.

Yet concerned with assistance during the workflow composition phase, Mohan, Ebrahimi & Lu (2015) proposes the use of Folksonomy (Gruber, 2007) to enrich the data used for the recommendation of others workflows similar to a workflow under development. A design workflow tool was developed that allows free specification tags to be used in each component, making it possible to use not only the recommendation strategy through the workflow syntax, but also component semantics. Soomro, Munir & McClatchey (2015) uses domain ontologies as a knowledge base to incorporate semantics into the recommendation process. A hybrid recommender system was developed using ontologies to improve the already known recommendation strategy based on the extraction of standards from other workflows. Zeng, He & Van der Aalst (2011) uses data and control dependencies between activities, stored in the workflow provenance to build a causality table and another weights table. Subsequently, a Petri network (Zhou & Venkatesh, 1999) is used to recommend other components for the composition of workflow.

In the context of helping less experienced users in the use of scientific workflows, Wickramarachchi et al. (2018) and Mallawaarachchi et al. (2018) show experiments that prove that SWfMS BioWorkflow (Welivita et al., 2018) use is effective in increasing student engagement and learning in Bioinformatics.

Some works propose recommendation approaches that assist less experienced users in analysis of unknown domains, as is the case of Kanchana et al. (2016) and Kanchana et al. (2017), where a chart recommendation system was developed and evolved based on the use of metadata from any domain data. The system uses Machine Learning and Rule-based components that are refined with user feedback on the usefulness of the recommended charts.

Most of the approaches that uses recommender system methods to support the scientific process are closely linked to the experiment’s composition phase. The execution phase, where there is a need to adjust parameters, still lacks alternatives. Table 6 compares related work with FReeP approach. In Table 6 we show the name of the approach (column Approach), if it is focused on a specific domain or if its generic (column Domain), if it prunes the search space or considers the entire search space (column Search Space), if the approach considers dependencies among parameters (column Considers Dependencies), if it requires a new execution of the workflow or the application (column Requires Execution), and in which phase of the experiment life-cycle the approach is executed (column Life-cycle Phase). If there is no information about the analyzed characteristics in the paper we set as N/A (Not Available) in Table 6. This work proposes a hybrid recommendation algorithm capable of making value recommendations for one or multiple parameters of a scientific workflow, taking into account the user’s preferences.

Table 6 Comparison between FReeP and related work.

Approach	Domain	Search space	Considers dependencies	Requires execution	Life-cycle phase	
Bergstra et al. (2011)	General	All	No	Yes	Execution	
Bergstra & Bengio (2012)	General	Pruned	No	Yes	Execution	
Eggensperger et al. (2013)	General	Pruned	No	Yes	Execution	
Zhang et al. (2016)	General	Pruned	No	Yes	Execution	
Gogna & Tayal (2013)	General	Pruned	No	Yes	Execution	
Zhou et al. (2018)	Workflow	N/A	No	No	Composition	
De Oliveira et al. (2008)	Workflow	N/A	No	No	Composition	
Mohan, Ebrahimi & Lu (2015)	Workflow	N/A	No	No	Composition	
Soomro, Munir & McClatchey (2015)	Workflow	N/A	No	No	Composition	
Zeng, He & Van der Aalst (2011)	Workflow	N/A	Yes	No	Composition	
Zhou & Venkatesh (1999)	Workflow	N/A	Yes	No	Composition	
Wickramarachchi et al. (2018)	Workflow	N/A	N/A	N/A	Composition/Execution	
Mallawaarachchi et al. (2018)	Workflow	N/A	N/A	N/A	Composition/Execution	
Kanchana et al. (2016)	Workflow	N/A	N/A	No	Analysis	
Kanchana et al. (2017)	Workflow	N/A	N/A	No	Analysis	
FReeP	Workflow	Pruned	Yes	No	Composition/Execution	

Final Remarks

The precision and recall results obtained from the experiments suggest that FReeP is useful in recommending missing parameter values, decreasing the probability that failures will abort scientific experiments performed in High-Performance Computing environments. These results show a high-reliability degree, especially in the recommendation for one workflow parameter due to the number of experimental iterations performed to obtain the evaluations. The low availability of data for the experiments of the recommendation for n parameters impacts the reliability of the results obtained in this scenario. However, the results presented for the n parameters recommendation show that the approach is promising.

FReeP has a number of characteristics pointing out its contribution in saving runtime and financial resources when executing scientific experiments. First, FReeP can be executed on standard hardware, such as that used in the experiments presented in this article, without the need for an HPC environment. Besides, FReeP does not require any further execution of the scientific workflow to assess the recommendation’s quality as it uses provenance data. This characteristic of not requiring an instance of the scientific experiment to be performed is the huge difference and advantage compared with Hyperparameter Optimization strategies widely used in the Machine Learning models tuning.

In FReeP, all training data are collected and each tuple represents a different execution of the workflow. This data gathering process can nevertheless be time-consuming. However, one aspect that is expected is that the recommendation process will be performed once and a series of executions of the same workflow is repeated a significant number of (varying the known parameter). In addition, in many research groups there is already a database containing the provenance (Freire et al., 2008) that can be used to recommend parameter values for non-expert users, i.e., the scientists will not need to effectively execute the workflow to train the model since provenance data is already available. Public provenance repositories such as ProvStore (https://openprovenance.org/store/) (Huynh & Moreau, 2015) can be used as input for FReeP. For example, ProvStore contains 1,136 documents (each one associated with a workflow execution) of several different real workflows uploaded by research groups around the world.

From the perspective of runtime, when using the ANOVA partitioning strategy, in the experimental evaluation with the provenance data from the Sciphy workflow, the average time spent on the recommendations is about only 4 minutes. In comparison, the average time of execution of the workflow Sciphy extracted from the provenance data used is about 17 h and 32 min. Still taking into account the use of the ANOVA partitioning strategy, in the experimental evaluation with the Montage workflow provenance data, the average time spent on the recommendation is about 1 h and 30 min. In contrast, the average execution time of a workflow experiment Montage extracted from the provenance data used was about 2 h and 3 min.

Although it is more evident the lower relation between the experiment’s execution time and the recommendation time when analyzing the data from the Sciphy workflow, it is essential to emphasize that more robust hardware is not necessary to execute the recommendation process. Yet, future improvements in FReeP includes employing parallelism techniques to further decrease the recommendation time.

Conclusion

The scientific process involves observing phenomena from different areas, formulating hypotheses, testing, and refining them. Arguably, this is an arduous job for the scientist in charge of the process. With the advances in computational resources, there is a growing concern about helping scientists in scientific experimentation. A significant step towards a more robust aid was the adoption of scientific workflows as a model for representing scientific experiments and Scientific Workflow Management Systems to support the management of experiment executions.

Computational execution of the experiments represented as scientific workflows relies on the use of computer programs that play the role of each stage of the experiment. In addition to input data, these programs often need additional configuration parameters to be adjusted to simulate the experiment’s conditions. The scientist responsible for the experiment ends up developing an intuition about the sets of parameters that lead to satisfactory results. However, another scientist who runs the same experiment will not have the same experience, which may lead him/her to define a set of parameters that will not result in a successful experiment.

Several proposals in the literature have aimed at supporting the composition phase of the experiments, but recommending parameter values for the experiment execution phase is still an open field. This article presented FReeP: Feature Recommender From Preferences, an algorithm for recommending values for parameters in scientific workflows considering the user’s preferences. The goal was to allow a new user to express their preferences of values for a subset of workflow parameters and recommend values for the parameters that had no preference defined. FReeP has three versions, all of them relying on Machine Learning techniques. Two approaches focused on the value recommendation for one parameter at a time. The third instance addresses recommending values for all the other parameters of a workflow for which a user preference was not defined.

The proposed algorithm proved to be useful for recommending one parameter, indicating a path for the recommendation of n parameters. Nevertheless, there are some limitations. FReeP, as a memory-based algorithm, faces scalability issues as its implementation can consume a lot of computational resources. Yet, the recommendations of FReeP are limited to the existence of examples on the provenance dataset. This means that the algorithm cannot make any “default” recommendations if there are no examples for the algorithm’s execution or recommend values that are not present in the provenance dataset. Also, the recommendation algorithm may have a longer processing time than the experiment itself. Another point is that all the instances have the same weight during the recommendation process. The algorithm does not consider the user’s expertise that performed the previous execution to adjust an example’s weight. Still, the algorithm considers only the set of parameters of the workflow; however, a set of parameters may be more or less relevant according to the input data. Additionally, the recommendation algorithm may end up recommending a set of values present in the provenance base that causes a workflow execution failure.

Based on those limitations, there are some proposals for future work. First proposal is parallelizing the processing of the generated partitions, which should decrease the time spent on the recommendation. In addition, evaluating FReeP on data from other domains and evaluating the tradeoff between the recommendation time and the algorithm execution time. Also, associating weights with examples from the provenance dataset according to the user’s profile. Lastly, using instances from the provenance dataset that failed to execute the workflow as a constraint to improve the recommendations’ results.

Supplemental Information

Supplemental Information 1 Provenance data extracted from actual executions of the Montage workflow.

One part was run using SciCumulus Workflow Management System, the other part retrieved from the Workflow Generator site (https://confluence.pegasus.isi.edu/display/pegasus/WorkflowGenerator).

The crota2 attribute is a float value that represents an image rotation on the sky. The dec attribute is an optional float representing Dec for region statistics, and crval2 is a float value representing Axis 2 sky reference value in Montage workflow.

Click here for additional data file.

Supplemental Information 2 Provenance data extracted from actual executions of the workflow SciPhy performed using SciCumulus Workflow Management System.

The attribute prob1 is the probability that a given evolutive relationship is valid. Prob2 attribute is the probability that a given evolutive relationship

is valid. The num aligns attribute is the total number of alignments in a given data file. Finally, the length attribute is the maximum sequence length in a specific data file.

Click here for additional data file.

The authors would like to thank Kary Ocaña for her explanations of the parameters of SciPhy workflow.

Additional Information and Declarations

Competing Interests

Author Contributions

Data Availability

1 Data sources are available at http://irsa.ipac.caltech.edu.

Daniel de Oliveira is an Academic Editor for PeerJ

Daniel Silva Junior conceived and designed the experiments, performed the experiments, analyzed the data, performed the computation work, prepared figures and/or tables, authored or reviewed drafts of the paper, and approved the final draft.

Esther Pacitti conceived and designed the experiments, authored or reviewed drafts of the paper, and approved the final draft.

Aline Paes conceived and designed the experiments, analyzed the data, authored or reviewed drafts of the paper, and approved the final draft.

Daniel de Oliveira conceived and designed the experiments, analyzed the data, authored or reviewed drafts of the paper, and approved the final draft.

The following information was supplied regarding data availability:

All the code and data are available at https://github.com/MeLL-UFF/FReeP.

The CSV files are available in the Supplemental Files.

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
