# Peer review of "Provenance-and machine learning-based recommendation of parameter values in scientific workflows"

_PeerJ Computer Science, doi:10.7717/peerj-cs.606_

## Round 0.1 · original submission · Major Revisions

The reviewers agree that the paper has merit. Some critical comments have been raised, especially from reviewer 2. The authors are invited to address them carefully.

·

Basic reporting

The paper has presented a parameter value recommendation method to suggest
values for workflow parameters based on user preferences.
The paper is well written and structured.

Experimental design

Result evaluation is sufficient.

Validity of the findings

It would be better to include a Discussion section, that compares the proposed methodology and results with the existing related studies. This will help to justify the validity and the novelty of the proposed study.

Additional comments

Few typos:
Equation (4) contains another numbering (1)

Name out the complete names of the parameter abbreviations using in the tables, with in the text description. (eg. Table 3, Table 4 - cnty, crval1)

Since the paper addresses a technology enhanced workflow management system, it would be better to consider latest related studies when comparing with the existing work. Since the paper will be published in 2021, better to consider latest related work/ applications (may be 5 years back)
For example, user feedback based rule-based recommendation systems were addressed in following references.
DOI: https://doi.org/10.1145/3018009.3018027
DOI: https://doi.org/10.1109/MERCon.2017.7980467
Some of the technology based workflow system studies in different domains can be found in
DOI= https://doi.org/10.1109/TNB.2018.2837122
DOI https://doi.org/10.3991/ijet.v13i12.8608
DOI: https://doi.org/10.1109/TALE.2018.8615134

Reviewer 2 ·

Basic reporting

The paper proposes FReeP: Feature Recommender from Preferences for recommending values for parameters in scientific workflows considering the user's preferences. The paper describes the recommendation strategies behind FReep, putting in context the challenges and burden to tune parameters when designing and running scientific workflows. Beyond the fact that tuning implies a heavy lifting task, it is also a time-consuming task. Moreover, it consumes resources that can lead to economic cost when the workflow is executed on "pay as you go" architectures like the cloud. The problem and challenges addressed are clearly stated by the paper together with a background on scientific workflows. The proposed recommendation approach has been experimentally validated using existing data sets. The focus is given on the precision and recall of the recommended parameters. Yet, the possibly reduced overhead between non-recommendation based tuning and recommendation based is not discussed. The recommendation task is just an automatic test and error process; it will finally find the "right parameters" for the workflow activities (precision and recall). But what about resources consumption? What about explicability and tracking of the choice of parameters? How can these benefits of recommendation be measured? The related work is too short and superficial. What about existing work that addresses parameters tuning for ML algorithms, of course, this is not a recommendation process. Still, the automatic search of "right" parameters behind is related to the addressed recommendation task. This is deceiving. The paper also lacks concrete examples of scientific workflows (e.g., a use case) across the "theory" descriptions, particularly in the experiments section.

Experimental design

The paper experiments are designed to evaluate precision and recall of three recommendation cases for which the work proposes a solution: Discrete Domain, Discrete and Continuous Domain, Generic FReeP Recommendation Evaluation. They are done using provenance data sets and specific machine learning and AI algorithms as a reference that are often used in scientific workflows. The recommendation techniques are assessed concerning precision and recall metrics. The paper describes reasonably the experiment setting and the results. The experiment does not show a concrete scientific workflow that might include provenance data from different activities that might use different algorithms. The parameters of a given algorithm might depend on tunning the parameters of another activity's parameters in the workflow. In this sense, I find the experiments too ideal and too synthetic. What about resource consumption, does recommendation help save resource consumption when a workflow must be executed many times? What about the cost of recommendation in terms of resources? Is it shorter than the repetitive execution of a scientific workflow? This kind of assessment must be done to complete experiments and validate FReeP.

Validity of the findings

There is no doubt about the interest of FReeP for the design and execution of scientific workflows involving tasks that need important tunning. Learning from previous executions using provenance data that are already collected is smart and strategic. The different strategies behind the system correspond to the type of parameters tuning associated with many analytics algorithms. It seems that scientific workflows might use and combine different such algorithms across their tasks; the question is whether tasks are independent or whether the recommendation might change when analysing all the tasks of a workflow. Are the user preferences enough input for dealing with the recommendation? Or do the characteristics of data impact the parameters' choice and, therefore, on the possible ways of addressing recommendation? Should recommendation not be interactive and user preferences are chosen or pondered by a scientist to guide recommendation?

Additional comments

The paper is a fair description of a recommendation solution for designing and tunning analytic activities composing scientific workflows. The strategy of using provenance data and exploiting them for guiding recommendation is smart. The paper needs the following modifications to consolidate it.
1. A motivation example of a scientific workflow can illustrate the problem and the proposed recommendation solutions. Of course, the audience targeted by the paper is one that knows what a scientific workflow is yet there are types and families of such workflows that might use different analytics algorithms with other parameters tuning challenges. It is essential to clarify and put in perspective this aspect in the explanations and the description of the global recommendation provided by FReeP.
2. Background. Sometimes the introduction of concepts focusses on a specific system or approach. For a background section, I suggest a more objective approach, either explaining why are chosen references enough for introducing concepts or selecting more references and synthesising different definitions and perspectives to build the section. The background or another section can give a taxonomy of scientific workflows and those that can best benefit specific parameters' recommendation. If parameters recommendation regards specific families of ML and AI, it must be clarified. I believe that different types of algorithms lead to additional provenance data and introduce other recommendation challenges. This type of discussion is missing in the paper.
3. The paper is motivated by the fact that scientific workflow tuning requires running the workflows several times, which is time and resources consuming. Yet, the remainder of the paper and particularly experiments do not longer respond to this kind of "research questions". If the paper's purpose is not intended to answer these questions and do not motivate it with them or precise the "research questions" or aspects of the problem that you answer in it. In this sense, I also believe that precision and recall assess the recommended parameters by the three proposed strategies. It seems that the recommender automates the test-error process when designing a scientific workflow, so it is not surprising that it works. The question is how much overhead it creates, and whether it consumes fewer resources than repeating the execution of a workflow to tune parameters.
4. Related work needs to be completed; it is too short and too general.

·

Basic reporting

**Clear and unambiguous, professional English used throughout.**

The article is reasonably well written.

**Literature references, sufficient field background/context provided.**

I am not an expert in the area, but I can't think of any references that are missing. If anything, there is an overabundance of background. The authors go into fine-grained detail on a number of topics, such as "SciCumulus" (lines 149-173) and collaborative filtering (lines 242-306) without providing any explanation for how these topics relate to their own work. For example, the authors devote over a page of their manuscript to explaining collaborative filtering, but don't explain how it relates to their method. The authors mention that this manuscript is a submission of a previously published conference paper which contained less background. If they feel that this level of detail is necessary to contextualize their contribution, they should make explicit references to how their model differs from the background. For example, "Collaborative filtering is a common recommendation paradigm... [~2 sentences of detail]. We do not use this method because X,Y,Z,. Instead, we contribute A,B,C"

**Professional article structure, figures, tables. Raw data shared.**

Figures 5,6,7,8,9,10 were very helpful and effectively communicated the method. Figure 11 should be improve (axes labels are too small, both heatmaps should use the same color scale). I recommend that the authors report their results to fewer significant figures in Tables 3 and 4.

The example in Table 1 is confusing. It might be easier to show four or five sample ballots, rather than the number of times a given ordering occured.

The authors provide a link to a GitHub repository with their code, but I can't find the provenance datasets.

**Self-contained with relevant results to hypotheses.**

Yes.

**Formal results should include clear definitions of all terms and theorems, and detailed proofs.**

N/A

Experimental design

## Experimental Design

**Original primary research within Aims and Scope of the journal.**

Yes.

**Research question well defined, relevant & meaningful. It is stated how research fills an identified knowledge gap.**

After reading the paper I am still confused about the core research question. I understand that the authors intend to recommend appropriate hyperparameters in scientific workflows, but to what end? Is the intent to recommend parameters that don't cause the workflow to crash (as line 82 suggests) or parameters that fit the users' prior preferences (as the experiments and name of the method imply)? Without this clarification I can't evaluate how meaningful the research question is.

**Rigorous investigation performed to a high technical & ethical standard.**

Yes.

**Methods described with sufficient detail & information to replicate.**

Yes.

Validity of the findings

## Validity of the Findings

**Impact and novelty not assessed. Negative/inconclusive results accepted. Meaningful replication encouraged where rationale & benefit to literature is clearly stated.**

The findings are important and could be impactful to the scientific community. However, the aforementioned ambiguity around the paper's core research questions makes a more detailed evaluation of contributions difficult.

**All underlying data have been provided; they are robust, statistically sound, & controlled.**

The authors should add more detail about how their data was collected. Are all of these runs from the same user, or multiple users? If they are from the same user, then the paper loses some claim to generalizability, since there is no evidence that the model is robust to parameters drawn from different user distributions. If they are from different users, then were validations folds split by user, or are samples from the sample user distributed across folds?

**Conclusions are well stated, linked to original research question & limited to supporting results.**

The conclusions could be better organized. I'd like to see a summary table or figure that lists the experiments the authors conducted and the key results. As is it's prohibitively difficult to scan the paper.

**Speculation is welcome, but should be identified as such.**

N/A

Additional comments

Thank you for your submission! I think supporting that scientific computing doesn't get enough attention from the computer science community and I'm glad to see (to me) new work in this space. I'd recommend that you:
1) Clarify the intent of your paper, and how it's supported by your experiments
2) Dramatically trim down your background, or better place it in the context of your paper
3) Better organize and highlight your results so that they can be digested in a quick read
I have attached an annotated manuscript. Please excuse my handwriting :)

---

## Round 0.2 · Minor Revisions

Please carefully address all final comments by reviewers before submission.

·

Basic reporting

It would be good, if the authors include some comparison tables (summary tables) of the related work, otherwise, it may become difficult to flow the textual description clearly.

Proof-read the paper for the improvements of writing comprehension.

The paper is improved and can be published.

Experimental design

Good

Validity of the findings

Sounds good.

Additional comments

It would be good, if the authors include some comparison tables (summary tables) of the related work, otherwise, it may become difficult to flow the textual description clearly.

Proof-read the paper for the improvements of writing comprehension.

The paper is improved and can be published.

---

## Round 0.3 · accepted · Accept

The authors carefully revised the paper.